# Decision Analysis on Sustainable Value: Comparison of the London and Taiwan Markets for Product Integration of Family Security Services and Residential Fire Insurance

**Jen-Chieh Lee [1,*] and Tyrone T. Lin [2]**

[1] Department of Risk Management and Wealth Planning, Takming University of Science and Technology, 56, Sec. 1, Huanshan Rd., Neihu, Taipei 11451, Taiwan

[2] Department of International Business, National Dong Hwa University, 1, Sec. 2, Da-Hsueh Rd., Shou-Feng Hualien 97401, Taiwan; tjlin@gms.ndhu.edu.tw

[*] Correspondence: leesir@takming.edu.tw

**Abstract:** This paper explores a decision analysis on product integration of family security services and residential fire insurance in the London and Taiwan markets by using the proposed mathematical models for counting sustainable value. This paper shows the five main different results between London and Taiwan markets with ten different parameters of the family security market, to find out the optimal number of family security integrated services for each security company in London. The improvement of the risk aversion effect based on risk and financial management will enhance the market share of the private security industries in the London and Taiwan markets. The results of this research can serve as a reference for the decision-making of private security industries on product integration under sustainable value consideration. The research findings highlight the potential benefits for both the private security industry and the insurance industry in their design and negotiation for product integration to improve both of business operation and achieve corporate social responsibility goals to match the sustainability in the future.

**Keywords:** sustainability; decision analysis; family security services; residential fire insurance; risk aversion

## 1. Introduction

This paper revises and extends the decision model based on the author's empirical research of the Taiwan security market (see Lee 2017) to focus on two major decision variables ($N$ and $d$), i.e., the optimal customers of the economic scale ($N$) and the optimal premium discount ($d$), to construct a decision model on risk aversion for the purpose of reducing the risks and enhancing the sustainability, estimating the premiums discount, and increasing the market share of family security services. Puelz (2010) proposed the technical effect of fire insurance in property and casualty insurance operations, but also revealed the function of guiding loss prevention, which would help consumers avert risk. However, insurance and security industries used to compete with each other, with customers buying family security services or residential fire insurance separately. This study proposes that they should work together to create mutually beneficial tactics in view of cooperative-competition theory and the real-life situations based on risk and financial management.

As the improvement of financial performance in the security service industries depend on the normal expansion of business, the security industry should promote decision analysis applications to deal with the uncertainty of consumer demand going forward. It is foreseeable that the competition will intensify with competitors from other sectors. Kim et al. (2013) found that academic literature has emphasized cooperation between channel members because of the interdependence between them; in reality, retailers may accept competition as just another part of doing business with suppliers. This paper viewed the family insurance services as retailers, and residential fire insurance as suppliers on product integration, not only to enhance the effect of consumer's risk aversion, but also intensify the competitiveness of the industry. This paper aims to examine the decision analysis pertaining to the integration of the residential security products and fire insurance policies to meet the demand from risk-averse consumers. The purpose is to calculate and validate the appropriate discount of insurance premiums that insurance companies provide to security companies for the combined product offerings. This is followed with the estimates of the market share for the residential security industry with the enhanced benefits to risk-averse consumers by combining risk control into the fully integrated services.

This paper specifically explains that the discount rate ( $d$ ) is not the capital cost rate or return on investment defined under general actual business activities, but the modified discount rate considering the perpetual operating value, which will be adjusted, varying with operating costs of the relevant industries. The increase of the written premiums for the residential fire insurance can also be estimated and compared with the different decision analysis results between the Taiwan and London security markets from the viewpoint of sustainable value. The London market is compared with the Taiwan market because Taiwan's security market is developing rapidly, and the experience of the London security market in the UK is quite mature and worth learning. Therefore, the author uses the model mentioned before, alongside practical data, to illustrate this comparison, mainly to indicate the feasibility of this model and to verify its consistency and the differences of the results in a multinational market. Another reason is that the London market is the most representative one in the world in terms of its scale or legal system. Its market information is relatively complete. Thus, it is worthy to use the London market as a contrast group for the model's development.

## 2. Literature Review

Diphoorn and Kyed (2016) pointed out that the privatization of the security industry in African cities is reflective of its rapid growth, which is matched by the widespread involvement of civilian actors in neighborhood watches and self-organized policing groups. This paper responds to fire prevention and control policy advocacy of the Taipei City government by putting forward innovative ideas of product integration of the private security and insurance industries. Xin and Huang (2013), using scenario clusters of residential fire risk analysis, determined that fire risk management can enhance the safety level of buildings and reduce fire risks and damages. This paper shows that integrated products/services can win customers' trust and enhance the risk aversion effect based on risk and financial management, a specific measure of fire risk management. Sajjad et al. (2015) also proposed that results reveal that sustainability values of top management, a desire to reduce risk and stakeholder management, are salient motivators for the adoption of sustainable supply chain management. Zhang et al. (2015) explored an integrative framework on customer involvement and service firm internationalization performance. This paper takes a similar integrated approach in product integration, so that customers can understand the need for risk aversion and risk diversification on the internationalization of the underwriting capacity in insurance companies. Dimakos and Aas (2004), incorporating the contemporary experts' knowledge in relationships between risks (rather than the conventional wisdom), gave a 20% reduction in the total economic capital for a year time horizon. In the meantime, the integrated risk management service of product integration can save a 20% premium on the residential fire insurance. Ali (2015) proposed that Karachi city's major problems are a lack of protection by law enforcement agencies, people's low level of awareness regarding fire incidence, and finally, the lack of firefighting readiness and

government support. However, this paper proposes the concept of product integration by using the example of the Taipei Municipal Government's requirement for families to install fire alarms in 2017.

Jennings (2013) reviewed the literature on social, economic, and building stock characteristics as they relate to residential fire risk in urban neighborhoods. In comparison, this paper conducts an interdisciplinary study to explore the product integration of family security services and residential fire insurance on sustainability. Chernonog and Kogan (2014) examined the effect of risk aversion and proposed the creation of risk aversion effects by extended pricing across the supply chain according to the perspective of consumers.

However, this paper suggests that it is necessary to maintain the economic scale of the private security companies and to assess the market share of the family security service market. The insurance industry should adjust its marketing strategy with flexibility in premium discounts to pursue a win–win strategy to achieve corporate social responsibility goals. Outreville (2014) reviewed literature and empirical studies on the demand for insurance with consideration for the use of variables associated with relative risk aversion. Based on the product integration of family security and insurance, this paper proposes that the industry should strengthen competitiveness by leveraging its existing foundation. Meulbroek (2002) indicated the trend for integrated risk management and provided insight into integrated risk management for consumers. Bandaly et al. (2014) contributed to the development of an integrated approach to supply chain risk management using operational methods and financial instruments. Lee and Wilson (2013) proposed that data of residential burglary incidents and residential burglar alarm permit records be integrated with a geographical information system program to analyze a spatial impact of alarms on burglary at the city-wide level. In comparison, this paper constructs mathematical models to verify product integration of security services for fire protection and residential fires insurance. Murphy (2013) proposed the use of various social media tools (e.g., microblogs, podcasts, and networking sites) to respond to fire services, and the use of security technology to integrate product offerings from different industries. Cihan (2016) found that the Turkish private security is rapidly growing relative to that in other countries, although the majority of Turkish private security officers felt that the level of training they have received was sufficient. To improve the interaction of the private security industry and the insurance industry, this paper presents the concept of integrated products/services to enhance the service level of the private security industry. Centered on the idea of effectively enhancing sustainability, the product integration of family security services and residential fire insurances will be conducive to individual company development or industrial competitiveness.

Oliveira et al. (2015) created a method, called "decision making based on knowledge," to analyze the decision-making process at the forefront of innovation. This paper is based on the decision-making of risk aversion knowledge and the decision-making analysis of innovation. Weinholt and Granberg (2015) demonstrated how cost–benefit analyses can be used for evaluating effects of collaborations via integrated products/services of security industries, and illustrated the social benefits of fire safety and rescue services from family security officers. Oliver (2015) identified the conditions when consumers place an emphasis on specific relational behaviors in evaluating the product use experience. Specifically, trust, commitment, and expertise seem more important. This paper zooms in on product integration to enhance the sustainability, the professional services of family security and insurance companies, and commitments to win customers' trust. Numerical simulation is performed to increase the likelihood of the success of the final product integration support. Skar et al. (2016) explored the mechanisms and strategies of integrating unorganized volunteers in emergency response and examined the decision-making of product integration with quantitative analysis. Štrukelj et al. (2020) explored the content of that paper, which incorporates principles of systems thinking and should be understood in synergy with other dimensions (sustainability paths).

In summary, this paper aims to explore the uncertainty of revenues of private security and insurance industries and applies decision analysis and the decision model to enhance consumers' risk aversion in the framework of integrated products/services from the viewpoint of sustainable value. In the research article, the author specifically mentioned that the Taipei City government's

requirement of installing fire extinguishers in 2017 proves that they pay great attention to fire accidents. Whereas, due to urban–rural differences, other cities do not have this requirement. The author imports relevant data of the London and the Taiwan market to the basic model. In theory, the data of the London market is mainly used as a contrast group. While in practice, the feasibility of the model can also be verified in different markets and scales.

## 3. Model Construction

### 3.1. Model Equation of Risk Aversion

Schulte and Hallstedt (2018) investigated the dynamics and implications of society's sustainability transition from a company risk management perspective to identify current risk management practices and preconditions for sustainability integration. Consumers purchase residential fire insurance to mitigate residential fire risk and possible losses. Such risk transfer can be viewed as risk aversion. In fact, there are two positive effects of risk aversion: One is to take control of treatments, and the other is to reduce the degree of loss. The family security service is essentially risk control. This paper puts forward the product integration of the private security industry in the context of risk aversion from consumers and optimal loss reduction (see Hiebert 1989). The product integration of family security services (as the main product) and residential fire insurance products (as the add-on product) does not only protect customers, but also boosts the residential fire insurance written premium ($Y_F(N), Y_R(N)$). As security companies serve the purpose of risk aversion, they may require a premium discount ($d$) from insurance companies. The premium discount may also affect the market share of the family security service ($M_S(d)$). Huber and Nowotny (2020) found that risk aversion has a robust and statistically significant negative impact on willingness to migrate within countries as well as abroad. This paper first lists two main functions of the relationship between the two graphs, the risk aversion model on product integration of family security services and residential fire insurance on sustainability. The hypothetical functions are as follows:

#### 3.1.1. Risk Aversion of Family Security Services

First, the assumptions of this article are as follows:

(1) A security service contract implies a risk premium of residential fire insurance. The cost is calculated for the private security industry.

(2) The private security industry is concerned about the level of insurance premiums and premium discounts.

(3) The insurance coverage has its compensation limit.

(4) The insurance coverage referred to in this paper is limited to residential fire insurance.

(5) According to an overview of the general insurance industry in the UK (ABI 2017), the average of net written family foreign premiums income in 2018, 2017, and 2016 is 2.3 billion British pounds (GBP). Therefore, this paper assumes a starting value $Y_0$ at 2.3 billion British pounds. The study also takes into account the industrial practice in London. The income of net written family foreign premiums is about 3.6 billion British pounds ($N$) in 2016; in other words, it implies the optimal product integration number and 1000 households as one unit. The symbols $Y_F(N)$ and $Y_R(N)$ indicate whether the optimal number of customers for the integrated products has reached the economic scale ($N$).

(6) These definitions of available symbols are as follows:

Because the handling of risks involves the business revenue and expenditure results of two different industries, there are related factors involved in financial analyses with a single tool and with an integrated tool. In order to perform a complete comparison and discussion, there are more parameters to consider in this study than other general research articles. For the parameter estimations and calculations based on the industries' meeting consumers' varied aspects of needs for hedging, the author re-organized the parameters into seven categories under the same group function

to facilitate the review of this paper. In addition, a sensitivity analysis is conducted after the model construction to reduce the natural bias caused by importing too many parameters into the model.

The first, we group the symbol of related written premiums are as follows:

$Y_F(N)$: The amount of written premium on residential fire insurance before the residential insurance service reaches the economic scale.

$Y_R(N)$: The amount of written premium on residential fire insurance after the residential insurance service reaches.

$y_{f_0}$: The average of the net written family foreign premiums income in 2018, 2017, and 2016 is 2.3 billion British pounds, the starting value assumed by this paper.

$y_{f_1}$: The coefficient of the polynomial function, yet to reach the optimal product integration number.

$y_{f_2}$: The net written family foreign premium payments from private security companies, yet to reach the optimal product integration number.

$y_{r_1}$: The coefficient of the polynomial function, yet to reach the optimal product integration number.

$y_{r_2}$: The amount of net written family foreign premiums to grow three times, yet to reach the optimal product integration number.

$k_1$: The parameter of the polynomial function, yet to reach the optimal product integration number.

$k_2$: The parameter of the polynomial function, yet to reach the optimal product integration number.

$d$: The premium discount for the counted sustainable value.

The second, we group the symbol of related market share of family security services are as follows:

$M_s(d)$: The market share of family security services. This paper assumes the market share of the family security services in London is about $18\%$ in 2017. (MarketResearch.com 2017)

$m_{s_1}$: The coefficient of the polynomial function on market share of family security services.

$m_{s_2}$: The estimated market share of family security services when the premium discount is zero.

$k_3$: The parameter of the exponential function on market share of family security services.

The third, we group the symbol of related finance of Private security companies are as follows:

$R_s(N)$: The total annual income of the service fee in private security companies.

$L_s(N)$: The total amount of loss payouts of the private security companies.

$R_s^I(N)$: The total income of integrated products/services.

$P_I(N)$: The total amount of insurance premiums absorbed by private security companies.

$d(N)$: The function of the premium discount for sustainable value.

$L_S^I(N)$: The total amount of the loss payouts of the private security company.

$M_S^I(N)$: The disbursement cost of a one-time investment by the family security companies.

$r_{s_1}$: The coefficient of the polynomial function with the total annual income of the service fees in private security companies.

$k_4$: The parameter of the polynomial function with the total annual income of the service fees in private security companies.

$r_{s_2}$ : The coefficient of the power function with the total annual income of the service fees in private security companies.

$r_{s_3}$ : The total annual income of the service fees in a private security company, before the offering of integrated products/services.

$l_{s_1}$ : The coefficient of the polynomial function with the total amount of the loss of payments of the private security company.

$k_5$ : The parameter of the polynomial function with the total amount of the loss of payments of the private security company.

$l_{s_2}$ : The coefficient of the power function with the total amount of the loss of payments of the private security company.

$l_{s_3}$ : The coefficient of the power function with the total amount of the loss of payments of the private security company.

$l_{s_4}$ : The estimated total amount of the loss payouts of the private security company, about forty percent of annual incomes.

The fourth, we group the symbol of related finance of integrated services are as follows:

$r_{is_1}$ : The negative coefficient of the polynomial function with the total income of integrated products/services.

$k_6$ : The parameter of the polynomial function with the total income of integrated products/services.

$r_{is_2}$ : Indicates the coefficient of the power function with the total income of integrated products/services.

$r_{is_3}$ : The total annual income of the service fees in private security companies, before the offering of integrated products/services.

$p_{i_1}$ : The coefficient of the straight-line equation function with the total amount of insurance premiums absorbed by the private security companies.

$k_7$ : The parameter of the straight-line equation function with the total amount of insurance premiums absorbed by the private security companies.

$p_{i_2}$ : The estimated total amount of the loss payouts of the private security company, before product integration.

The fifth, we group the symbol of related premium discount of integrated services are as follows:

$d_1$ : The coefficient of the polynomial function with the premium discount, a negative value.

$k_8$ : The parameter of the quadratic polynomial function with the premium discount.

$d_2$ : The coefficient of the polynomial function with the premium discount, a negative value.

$d_3$ : The sum of the premium discount and commission, assumed to be 20% and 10%, respectively, in this paper.

The sixth, we group the symbol of related loss payouts of the security companies are as follows:

$l_{is_1}$ : The coefficient of the polynomial function with the total amount of the loss payouts of the security companies upon product integration.

$k_9$: The parameter of the polynomial function with the total amount of the loss payouts of the private security companies upon product integration.

$l_{is_2}$: The coefficient of the power function with the total amount of the loss payouts of the security companies upon product integration.

$l_{is_3}$: The total amount of the loss payouts of the private security company upon product integration.

The seventh, we group the symbol of one-time investment by the private security companies are as follows:

$m_{is_1}$: The coefficient of the exponential function of one-time investment in the security element of the integrated product. This paper assumes $m_{is_1} = 7.71$ (the unit is in one hundred British pounds) (Manta.com 2017).

$m_{is_2}$: The power of the exponential function on the disbursement cost of a one-time investment by the private security companies.

Equation (1) shows the relationship between the optimal number of customers for integrated products and the amount of net written family foreign premium on family insurance as follows:

$$Y_F(N) = y_{f_0} + y_{f_1} \times N^{k_1} + y_{f_2}, 0 \leq N < N_0;$$

$$Y_R(N) = Y_F + y_{r_1} \times N^{k_2} + y_{r_2}, N \geq N_0 \tag{1}$$

This paper presumes that the private security company is willing to give a gift with limited insurance coverage services for consumers, due to the time decentralized for financial risk of marketing practices and the increasing customers in the product integration. Additionally, private security companies can absorb small insurance premiums, and the additional insurance coverage also varies because of the size of the assets, the perceived risks, and cost considerations of the professional managers in the private security company. The initial written premium on residential fire insurance is increasing because of the increasing number of customers for integrated products offered by private security companies. Residential fire insurance premium income has been slowly increasing due to market competition; therefore, the product integration may meet the demand of risk-averse consumers and achieve corporate social responsibility goals. According to the calculus calculations shown in Equation (1), this paper assumes $Y_F'(N) = y_{f_1} \times k_1 \times N^{k_1-1} > 0, Y_R''(N) = y_{r_1} \times k_2 \times (k_2-1) \times N^{k_2-2} < 0; y_{f_2}$ and $y_{r_2}$ are the residential fire insurance premium incomes when $N=0$, and the result is $y_{f_1} > 0, y_{f_2} > 0, y_{r_1} < 0, y_{r_2} > 0$.

### 3.1.2. Risk Aversion of Residential Fire Insurance

The second, the assumptions of this article are as follows:

(1) Although the single residential fire insurance premium income is reduced because of premium discount ($d$), the claims cost will be also reduced due to the loss of control, and the overall written premium of residential fire insurance will increase.

(2) The greater the premium discount, the larger the market share of the family security service ($M_S(d)$). When $d$ is equal to zero, the market share of family security service is $m_{s_2}$. Based on the experiences of family security companies in London, this paper assumes that the market share of the family security service in London was about 18% in 2017 (MarketResearch.com 2017).

Equation (2) shows the relationship between family security service market share and premium discount as follows:

$$M_S(d) = m_{s_1} \times (1 - e^{-k3 \times d}) + m_{s_2} \tag{2}$$

Based on the different attributes of insurance and security, the decrease in the insurance premium is the result of the loss of control by the private security and its channels. The decreasing percentage is calculated from the premium discount coefficient, and its difference is also determined by the asset size of the insurance company to achieve corporate social responsibility goals, the increasing awareness of professional managers, and the risk control function of private security. The increase of the premium discount is conducive to the all-round marketing of the private security company, and the initial market share of the family security service will significantly increase. This is limited by the maximum discount rate of the premium incomes for insurance companies; thus, the market share of the family security service market will stabilize, and the product integration may meet the demand from risk-averse consumers based on risk and financial management. According to the calculus calculations in Equation (2), this paper assumes $M_s'(d) = m_{s_1} \times k_3 \times (1 - e^{-k_3 \times d}) > 0$, then $m_{s_1} > 0$; when $d = 0$, $m_{s_2}$ is the premium discount, then $m_{s_2} > 0$, the result is $m_{s_1} > 0, m_{s_2} > 0$.

### 3.2. Decision Model of Risk Aversion

This paper proposes many functions derived from the above model and refers to the net present value method ($NPV$) to assess the optimal product integration for risk-averse customers. This paper views the private security as a single service contract (usually three years), with the net present value ($\pi_1(N)$) equal to the total service fee of security company ($R_S(N)$) minus the total amount of damages ($L_S(N)$) by the private security. The net present value of product integration ($\pi_2(N)$) is equal to $R_S(N)$ minus the total amount of insurance premiums absorbed by a private security company $P_I(N)$ (multiplied by the premium discount $d(N)$) and minus the total amount of the loss of payments of the insurance ($L_S^I(N)$). Assuming the result remains the same, that is $\pi_1(N) = \pi_2(N)$. The arrangement Equation (3) is as follows:

$$R_S(N) - L_S(N) = R_S^I(N) - P_I(N) \times d(N) - L_S^I(N) \tag{3}$$

Equation (4) shows the relationship between the total annual income of the service fee and the optimal number of customers for the integrated products as follows:

$$R_S(N) = r_{s_1} \times N^{k_4} + r_{s_2} \times N^{k_4 - 1} + r_{s_3} \tag{4}$$

$R_S(N)$ is the total increase of the service fee of security of the company when the number of customers increases for the optimally integrated products. An analysis on the graph function depicting its first-order derivative suggests that $R_S'(N)$ is greater than zero. $R_S''(N)$ is also greater than zero with its second-order differential, which shows the company is willing to provide integrated products/services. According to calculus calculations, this paper assumes $R_S'(N) = r_{s_1} \times N^{k_4 - 1} + r_{s_2} \times N^{k_4 - 2} > 0$, $R_S''(N) = r_{s_1} \times (k_4 - 1) \times N^{k_4 - 2} + r_{s_2} \times (k_4 - 2) \times N^{k_4 - 3} > 0$, it gets $r_{s_1} > 0$, $r_{s_2} > 0$, $r_{s_3} > 0$.

Equation (5) shows the relationship between the total amount of loss payouts of the private security and the optimal number of customers for the integrated products as follows:

$$L_S(N) = l_{s_1} \times N^{k_5} + l_{s_2} \times N^{k_5 - 1} + l_{s_3} \times N^{k_5 - 2} + l_{s_4} \tag{5}$$

This paper assumes $L_S(N)$ (in ten million NT dollars) is the total amount of loss payouts by the private security industry (or companies), with initial increase in line with a higher number of customers. As the optimal number of customers for integrated products increase, the initial loss payouts of its private security also increase, but given the compensation limit of the maintenance

service contract, the first-order derivative $L_S'(N)$ is greater than zero, and the second-order differential $L_S''(N)$ is less than zero. According to the result of calculus calculations, this paper assumes

$$L_S'(N) = l_{s_1} \times k_5 \times N^{k_5-1} + l_{s_2} \times (k_5-1) \times N^{k_5-2} + l_{s_3} \times (k_5-2) \times N^{k_5-3} > 0,$$

$L_S''(N) = l_{s_1} \times k_5 \times (k_5-1) \times N^{k_5-2} + l_{s_2} \times (k_5-1) \times (k_5-2) \times N^{k_5-3} + l_{s_3} \times (k_5-2) \times (k_5-3) N^{k_5-4} < 0$ , it gets $l_{s_1} < 0, l_{s_2} > 0, l_{s_3} > 0, l_{s_4} > 0$.

Equation (6) shows the relationship between the total income of integrated products/services and the optimal number of customers for the integrated products as follows:

$$R_S^I(N) = r_{is_1} \times N^{k_6} + r_{is_2} \times N^{k_6-1} + r_{is_3} \tag{6}$$

This paper assumes $R_S^I(N)$ (in ten million NT Dollars) is the total income of integrated products/services, which will increase in line with a higher number of customers for the security companies in the oligopolistic market for family security services. The function of the first-order derivative $R_S^I(N)$ is greater than zero, and the second-order differential $R_S^{I''}(N)$ is less than zero. Based on calculus calculations, this paper assumes $R_S^{I'}(N) = r_{is_1} \times (k_6) \times N^{k_6-1} + r_{is_2} \times (k_6-1) \times N^{k_6-2} > 0$,

$R_S^{I''}(N) = r_{is_1} \times (k_6) \times (k_6-1) \times N^{k_6-2} + r_{is_2} \times (k_6-1) \times (k_6-2) \times N^{k_6-3} < 0$, it gets $r_{is_1} < 0$, $r_{is_2} > 0$, $r_{is_3} > 0$.

Equation (7) shows the relationship between the total amount of insurance premiums absorbed by private security companies and the optimal number of customers for the integrated products as follows:

$$P_I(N) = p_{i_1} \times N^{k_7} + p_{i_2} \tag{7}$$

This paper assumes $P_I(N)$ (in NT\$10 million) is the total amount of insurance premiums absorbed by private security companies for the combined product, and this amount will increase linearly in line with the growing number of customers of integrated products. This will help the insurance industry understand the premium absorption.

According to the calculus calculations shown in Equation (7), this paper assumes $P_I'(N) = p_{i_1} \times k_7 \times N^{k_7-1} > 0$ , then $p_{i_1} > 0$ ; when $N = 0$ , $p_{i_2}$ is the total amount of insurance premiums absorbed by private security companies. Therefore, when $p_{i_2} > 0$ , the result is $p_{i_1} > 0$ , $p_{i_2} > 0$ .

Equation (8) shows the relationship between the premium discount and the optimal number of customers for the integrated products as follows:

$$d(N) = d_1 \times N^{k_8} + d_2 \times N^{k_8-1} + d_3 \tag{8}$$

This paper presumes that, given the competitive dynamics of the private security market, the premium discount will be slowly reduced due to the limited cost of security and the lower risk of residential fire insurance. The first order differential $d'(N)$ is less than zero, and the second order differential $d''(N)$ is also less than zero. This suggests that the premium discount may affect the optimal number of customers for the integrated products. This helps the insurance companies to grasp how the discount in insurance premiums affects the number of their customers.

According to the calculus expressed in Equation (8), this paper assumes
$d'(N) = d_1 \times k_8 \times N^{k_8-1} + d_2 \times (k_8-1) \times N^{k_8-2} < 0,$

$d''(N) = d_1 \times k_8 \times (k_8-1) \times N^{k_8-2} + d_2 \times (k_8-1) \times (k_8-2) \times N^{k_8-3} < 0$, it gets $d_1 < 0$, $d_2 < 0$, $d_3 > 0$.

Equation (9) shows the relationship between the total amount of the loss payments by the insurance company and the optimal number of customers for the integrated products as follows:

$$L_S^I(N) = l_{is_1} \times N^{k_9} + l_{is_2} \times N^{k_9-1} + l_{is_3} \tag{9}$$

This paper assumes that $L_S^I(N)$ (in ten million yuan) is the total amount of the loss payments of the insurance, to be increased in line with the number of the optimally integrated products of security and insurance, and the loss of the family security by the integrated products/services shall be increased at the beginning, owing to limited payments by security companies. This will decrease ultimately, so the first-order derivative $L_S^{I'}(N)$ is greater than zero, and the second-order differential $L_S^{I''}(N)$ is less than zero. This suggests that the total amount of the loss payments of the insurance may be affected by the number of customers for the optimally integrated products of security and insurance. Through available mathematical calculus, in practice, this helps to understand how the damages for loss paid by the security companies for the integrated products affects the number of customers for the security companies. According to the calculus expressed in Equation (9), this paper assumes $L_S^{I'}(N) = l_{is1} \times k_9 \times N^{k_9-1} + l_{is2} \times (k_9-1) \times N^{k_9-2} > 0$,

$L_S^{I''}(N) = l_{is1} \times k_9 \times (k_9-1) \times N^{k_9-2} + l_{is2} \times (k_9-1) \times (k_9-2) \times N^{k_9-3} < 0$; the result is $l_{is_1} > 0$, $l_{is_2} > 0$, $l_{is_3} > 0$.

Equation (10) shows the relationship between disbursement cost of one-time investment by the family security industry and the optimal number of customers for the integrated products as follows:

$$M_S^I(N) = m_{is_1} \times (1 - e^{-m_{is_2} \times N}) \tag{10}$$

This paper proposes that $M_S^I(N)$ is a disbursement cost of a one-time investment by the family security (in ten millions NT dollars), and the cost is increased along with the rise in the optimal number of customers for the integrated products, given the private security industry's reliance on human capital and technology. As the average fixed costs can be reduced, the first-order derivative $M_S^{I'}(N)$ is greater than zero, and the second-order differential $M_S^{I''}(N)$ is less than zero. This suggests that the disbursement cost of a one-time investment by the family security industry may be affected by the number of customers for the optimally integrated products of security and insurance. Through available mathematical calculus, this paper assumes $M_S^{I'}(N) = m_{is_1} \times m_{is_2} \times (1 - e^{-m_{is_2} \times N}) > 0$,

$M_S^{I''}(N) = -m_{is_1} \times (m_{is_2})^2 \times (1 - e^{-m_{is_2} \times N}) < 0$; it gets $m_{is_1} > 0$, $m_{is_2} > 0$.

Equation (11) shows the financial balance of the product integration in a security company as follows:

$$R_S(N) - L_S(N) + M_S^I(N) = R_S^I(N) - P_I(N) \times d(N) - L_S^I(N) \tag{11}$$

This paper presumes private security companies want to maintain the integrated marketing to meet the demand from risk-averse consumers and the sustainable value, which should be consistent with the company's financial balance of payments, where $\pi_s(N)$ denotes the average total loss payouts for compensation, plus the disbursement cost of a one-time investment by the family security industry ($M_S^I(N)$). This is at least equal to the average overall family security fee income ($\pi_j(N)$), using the decision-making model to assess the number of customers for the optimally integrated products of security and insurance. It should be consistent with the appropriate premium discount to meet the demand for integrated products from risk-averse consumers. $m_{is_1}$ (in the unit of one hundred British pounds) denotes the disbursement cost of a one-time investment by each household. This paper gets the equal function (11) integrated above as following: $\pi_s(N) + M_S^I(N) = \pi_j(N)$.

This paper constructs a decision model on the integration of both services into a single product to achieve corporate social responsibility goals, in order to identify the optimal number of family security services. It is based on two main variables, i.e., the optimum number of customers for integrated products ($N$) and the premium discount for sustainable value ($d$). This paper also

assumes that the market share of family security services and the written premium of residential fire insurance can be estimated, and the sensitivity of decision variables ($N$ and $d$ ) will concern the business development of both industries. It is hoped that the research findings will provide an adequate basis for decision-making of the private security industry, and may shed light on the terms of the contract between the private security industry and the insurance industry, to enhance the business of the former. Enterprises can consider the risks and financial burdens of consumers through systematic thinking, and it will be able to achieve the goal of sustainable operation.

## 4. Numerical Simulation Analysis

Rahman et al. (2017) presented a risk management approach for a sustainable cloud migration, considering four dimensions of sustainability, i.e., economic, environmental, social, and technology to determine the viability of cloud for the business context. The risks are systematically identified and analyzed based on the existing in-house controls and the cloud service provider offerings. This paper defines the variable in the mathematical model mentioned above, with several reasonable parameters for simulation that conform to the function of those equations. The risk aversion model based on risk and financial management for the product integration of family security services and residential fire insurance is substituted with analog values according to the function Equations (1) to (10). The ten independent variables specific to the London security market in Table 1, and the twenty-six coefficient parameters remainders, are consistent with the author's research models in 2017. Those parameters are as follows:

$$y_{f_0} = 2.3, y_{f_1} = 0.15, k_1 = 3, y_{f_2} = 0.12, y_{r_1} = -0.01, k_2 = 3, y_{r_2} = 0.36, m_{s_1} = 0.6, k_2 = 0.5,$$

$$m_{s_2} = 0.18, r_{s_1} = 0.08, k_4 = 2, r_{s_2} = 0.01, r_{s_2} = 0.01, r_{s_3} = 0.6, l_{s_1} = -0.001, k_5 = 3, l_{s_2} = 4,$$

$$l_{s_3} = 2, l_{s_4} = 0.24, r_{is_1} = -0.2, k_6 = 0.5, r_{is_2} = 0.1, l_{s_2} = 4, r_{is_3} = 0.6, p_{i_1} = 0.05, k_7 = 1, p_{i_2} = 0.2,$$

$$d_1 = -0.002, k_8 = 2, d_2 = -0.05, d_3 = 0.3, l_{is_1} = 1.6, k_9 = 0.5, l_{is_2} = 1.2, l_{is_3} = 0.27, m_{is_1} = 7.71, m_{is_2} = 1.5.$$

The numerical analysis of the POLYMATH values of Equation (11) derives variable values of the nonlinear equation calculation $N$. The assumptions for $36$ independent variables to assume the parameter values are estimated based on 10 dependent variables. The optimal number of customers for the integrated products can be estimated that there are about $1349$ ( $N = 1.349 \times 1000$ ); that is, if the economic scale is to be achieved and the private security industry is to become more competitive, N will be set into Equation (8). The calculated appropriate premium discount $d = 0.2288892$, or $22.89\%$, can be used as a risk control and channel distribution of the product integration of family security services, and $d = 0.2288892$ will be set into Equation (2). The market share of the family security service is estimated to reach $24.49\%$ ( $M_s(d) = 0.2448832$ ), and $N$ may be set into Equation (1). This paper estimates the amount of written premiums on residential fire insurance in the end of 2018, that is, $Y_F(N)$ to be approximately $2.79$ billion British pounds and $Y_R(N)$ to be $3.12$ billion British pounds.

The different parameters between the Taiwan market and the London market are as follows:

**Table 1.** The different parameters between the Taiwan and London security markets.

| Independent Variable | $y_{f_2}$ | $y_{r_2}$ | $m_{s_2}$ | $r_{s_3}$ | $r_{is_3}$ | $d_3$ | $l_{is_3}$ | $m_{is_1}$ | $l_{s_4}$ | $y_{f_0}$ |
|---|---|---|---|---|---|---|---|---|---|---|
| Taiwan [a] | 0.1 | 0.3 | 0.03 | 2.4 | 2.4 | 0.369 | 1.08 | 5.0 | 0.96 | 9.2 |
| London | 0.12 | 0.36 | 0.18 | 0.6 | 0.6 | 0.3 | 0.27 | 7.71 | 0.24 | 2.3 |

Note: [a] Lee (2017), Studies on Optimal Decision Models for Integrating the Products of Family Security Services and Residential Fire Insurances.

The sensitivity analysis compares the decision variables $d$ and $N$, which have a $30\%$ up and down variation, derived from the simulation analysis under a rationale normative. POLYMATH

calculation using Equations (2) and (8) derives the changing of the $M_S(d)$ as $-0.0803 \sim 0.0739$ via the changing of $d$; the changing of $Y_F(N)$ is $-0.0869 \sim 0.1582$ via the changing of $N$; the changing of $Y_R(N)$ $-0.0723 \sim 0.1318$ via the changing of $N$. The table of sensitivity analysis results in the London market is as follows:

According to Results of sensitivity analysis with decision variables in the London market Table 2, when $d$ is increasing by $30\%$, the greater the premium discounts offered by insurance companies to private security company, the more beneficial to the private security company, and $M_S(d)$ will increase by about $26\%$. In contrast, when $d$ is decreasing by $30\%$, the cut in premiums discount will not be able to boost the market share of the private security company. In fact, the share will be reduced about $8\%$. When the number of $N$ is increasing by $30\%$, $Y_F(N)$ can be increased by $15.82\%$. If $N$ is decreasing by $30\%$, $Y_F(N)$ will be reduced by $8.69\%$, and the same as $Y_R(N)$ can be increased by $13.18\%$ and reduced by $7.23\%$.

**Table 2.** Results of sensitivity analysis with decision variables in the London market.

| Decision Variables | Up and Down 30% | $M_S(d)$ | Up and Down | $Y_F(N)$ | Up and Down | $Y_R(N)$ | Up and Down |
|---|---|---|---|---|---|---|---|
|  | 0.2976 | 0.2630 | 0.0739 |  |  |  |  |
| $d$ | 0.2289 | 0.2449 |  |  |  |  |  |
|  | 0.1602 | 0.2267 | −0.0803 |  |  |  |  |
|  | 1.7542 |  |  | 3.2297 | 0.1582 | 3.5357 | 0.1318 |
| $N$ | 1.3494 |  |  | 2.7886 |  | 3.1240 |  |
| (unit: thousand) | 0.9446 |  |  | 2.5464 | −0.0869 | 2.8980 | −0.0723 |

Note: POLYMATH calculation based on Equations (2) and (8).

Therefore, for the product integration of family security services and residential fire insurance to avoid the risk associated with numerical simulation analysis, the POLYMATH results of decision-making can serve as a basis for the determination of the economic scale for the optimal number of customers to a private security company and sustainability. An appropriate premium discount can be calculated; that is, the insurance company should be able to give the private security company a discount percentage, so that the product integration of family security services and residential fire insurance will enhance the added value offered to risk-averse customers.

This paper quotes the author's doctoral dissertation "Product integration of family security services and residential fire insurance under risk aversion model" and attempts to update the empirical data from the Taiwan market and the London market. Ten parameters of the family security market in Taiwan and London are estimated based on experience values.

In order to demonstrate the stability of the risk aversion model for sustainability, the 36 original variables of the function coefficients remain unchanged, and only the 10 empirical values of the London market are modified. This paper compares the different empirical values between the Taiwan and London security markets. This paper also shows the key different results between the Taiwan and London markets as follows, see Table 3:

**Table 3.** Key differences in findings between the Taiwan and London security markets.

| Dependent Variable | $Y_F(N)$ | $Y_R(N)$ | $N$ | $d$ | $M_S(d)$ |
|---|---|---|---|---|---|
| Taiwan [b] | 9.51498 | 9.80065 | 1.127 | 0.3100844 | 0.1161726 |
| London | 2.78855 | 3.12398 | 1.349 | 0.2288892 | 0.2448832 |

Note: [b] Lee (2017), Studies on Optimal Decision Models for Integrating the Products of Family Security Services and Residential Fire Insurances.

This paper imports the parameter values from the London security market. The calculation results in $10$ dependent variables due to $36$ independent variables, and 5 main variables are used as the basis for the estimates of the possible development of the security market in London. This paper lists and analyzes the five main variables, comparing the differences between the Taiwan and London security markets. Property insurance industries in Taiwan may boost fire written premiums from NT\$9.2 billion to NT\$9.515 billion, and eventually up to NT\$9.801 billion. The average net written family foreign premiums income during the recent three years of 2018, 2017, and 2016 in the UK was 2.3 billion British pounds, which may increase to 2.789 billion British pounds, and up to 3.124 billion British pounds. The optimal family security customers for each company are $1127$ customers in Taiwan, and the optimal in London are $1349$, based on the product integration model. If the fire insurance premiums can be reduced by 22%, the market share of family security in London will increase to $24\%$ (currently about 18%). Supposing the fire insurance premiums can give $32\%$ in Taiwan, it will increase the security market share by nearly $12\%$.

According to the comparison of family security market shares going forward as predicted, London's security industry will remain double the market share of Taiwan's family security industry in the future. This is due to the relatively mature nature of the London security market. These findings may also offer insight to property insurance companies in the London and Taiwan markets, and prove the feasibility of product integration of family security services and residential fire insurance on sustainability. It obtains the results of the sustainable value of products with better risk aversion through data calculation and analysis of cross-industry combination.

## 5. Conclusions

Hoogendoorn et al. (2019) argued that sustainable entrepreneurs face specific challenges when establishing their businesses owing to the discrepancy between the creation and appropriation of private value and social value. The contribution of this research paper is based on the decision analysis model developed with empirical evidence in the Taiwan and London security markets and numerical simulation on the integration of family security services and residual fire insurance products to enhance value delivered to risk-averse customers and sustainability.

### 5.1. Academic Implication

In view of the product-service system as a new model, the integration of goods and services provides customers with added sustainable value and lays the cornerstone of competitive strategy (See Phumbua and Tjahjono 2012). This paper presents the five main different results between Taiwan and London with ten different parameters of the family security market to identify the optimal number of family security companies offering integrated service. In response to consumers' needs of integrated hedging, it is clarified in this study that the feasibility of specific cooperation policies between the insurance and the security industries can be verified through mathematical model establishments and reasonable data analyses. By importing actual data, estimated results are generated, which can be used as a reference for decision-making in business marketing to achieve sustainable value operation.

### 5.2. Management Implications

This paper proposes the product integration of family security services and residential fire insurance based on risk and financial management. In this paper, the sensitivity analysis of the decision variables indicates the probable business development of both industries. The decision-making algorithm has management implications. The improvement of the risk aversion effect in the decision-making process of product integration for the private security and insurance industries will enhance the competitiveness of the private security companies and increase the likelihood of premium income of residential fire insurance under the sustainable value concept. Sensitivity analysis results are shown as Table 2. The research findings can serve as a reference for the product integration decision-making of the private security industry to match sustainability in the future. It

can also benefit both the private security industry and the insurance industry to achieve corporate social responsibility goals in the contract reviewing process, particularly for the London security market. This paper posits that London's security companies will see double the market share compared to their counterparts in Taiwan due to the relative maturity of the London security market.

### 5.3. Research Contributions

This paper views residential fire insurance companies as suppliers and family security service companies as retailers for the product integration to cater to risk-averse customers with the sustainable value consideration. The contribution of this paper lies in the data calculation and analysis of cross-industry combination to obtain the results of the sustainable value of products with better risk aversion, and the enterprise can consider the risks and financial burdens of consumers through systematic thinking, and it will be able to achieve the goal of sustainable operation with product competitiveness.

Based on practical experiences in insurance and security services, the author establishes a mathematical model to obtain a reasonable discount result, which can be used as a reference when two parties cooperate and negotiate with each other. This article is about the application of sustainable value in the service industry. The consumer-integration services provided via a cooperation between insurance and security service companies will meet consumers' more comprehensive risk and financial planning, and will enhance the insurance and security service companies' business market share. This will be verifiable in practice. The integration mechanism of the insurance and the security industries can be applied to online alliance marketing services, allowing consumers to shop online through the Internet of Things. In this way, sustainable industrial operation could increase discounts on the items sold online to reciprocate consumers.

### 5.4. Research Limitation

This paper discusses the decision analysis on the sustainable value of cross-industry integration. Because there are many projects in the business operation, this article only focuses on the residential fire by the related industries; because it is not easy to obtain the practical information from related industries, this paper can only obtain statistical data through the relevant mechanism, so there will be a time gap; in addition, not limited to the scope of the research, future researchers can analyze commercial fires by the insurance industries and the security industries to facilitate the business integration by the cross-industry again.

**Author Contributions:** Conceptualization, T.T.L. and J.C.-L.; methodology, T.T.L.; software T.T.L.; validation, T.T.L. and J.C.-L.; formal analysis, J.C.-L.; investigation, J.C.-L.; resources, T.T.L.; data curation, J.C.-L.; writing—original draft preparation, J.C.-L.; writing—reviewrone T.T.L. and editing, J.C.-L.; visualization, T.T.L.; supervision, T.T.L.; project administration, T.T.L.; funding acquisition, T.T.L. All authors have read and agreed to the published version of the manuscript.

**Funding:** The authors would like to thank the Ministry of Science and Technology of the Republic of China, Taiwan for financially supporting this research under Contract No. MOST 107-2410-H-259-013-.

**Conflicts of Interest:** The authors declare no conflict of interest.

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
