# Peer review of "Decision Analysis on Sustainable Value: Comparison of the London and Taiwan Markets for Product Integration of Family Security Services and Residential Fire Insurance"

_jrfm, doi:10.3390/jrfm13110266_

Round 1
Reviewer 1 Report
The paper advances a decision analysis model with empirical evidence and numerical simulations on product integration of family security services and residential fire insurance in the London and Taiwan. I acknowledge the amount of work invested in this paper, therefore, overall, I consider it would contribute to the literature.
I would however suggest to enhance the innovative side of the paper, particularly the introduction and the conclusions sections should be reconfigured to rather encompass how this paper is different from other studies in this scientific field and to clearly state the mechanisms and specific policies through which own findings can serve as reference for decision making, not only simply stating this. Especially, in the conclusions section, instead of restating what the authors did in their paper, which is already mentioned by that section too many times, the authors should start a debate on the results (a section which is practically missing) and the subsections of academic and managerial implications, as well as research contributions should be significantly enhanced.
Author Response
Thank you for your opinions (reviewer 1), the author responded as follows:
The paper advances a decision analysis model with empirical evidence and numerical simulations on product integration of family security services and residential fire insurance in the London and Taiwan. I acknowledge the amount of work invested in this paper, therefore, overall, I consider it would contribute to the literature.
I would however suggest to enhance the innovative side of the paper, particularly the introduction and the conclusions sections should be reconfigured to rather encompass how this paper is different from other studies in this scientific field and to clearly state the mechanisms and specific policies through which own findings can serve as reference for decision making, not only simply stating this. Especially, in the conclusions section, instead of restating what the authors did in their paper, which is already mentioned by that section too many times, the authors should start a debate on the results (a section which is practically missing) and the subsections of academic and managerial implications, as well as research contributions should be significantly enhanced.
- I would however suggest to enhance the innovative side of the paper, particularly the introduction and the conclusions sections should be reconfigured to rather encompass how this paper is different from other studies in this scientific field.
Response:
The author will reconfigure the introduction and conclusion sections of this paper: The introduction section follows the order of the keywords of this paper. The repetitive content in the paper will be removed and substituted by a discussion of its results (the missing ones) and sub-parts. Besides, the differences between this study and research papers in other scientific fields will be explained explicitly, including the establishment of a reasonable mathematical model, integrated risk management, alliance service marketing, financial decision analysis, and cross-country comparative research. This is an innovative study built on many theories.
(P1-2)
This paper revises and extends the decision model based on the author’s empirical research of the Taiwan security market (see Lee 2017) to focus on two major decision variables ( and ), i.e. the optimal customers of the economic scale ( ) and the optimal premium discount ( ), to construct a decision model on risk aversion for the purpose of reducing the risks and enhancing the sustainability, estimating the premiums discount, and increasing the market share of family security services. Puelz (2010) proposed the technical effect of fire insurance in property & casualty insurance operations, but also revealed the function of guiding loss prevention, which would help consumers avert risk. However, insurance and security industries used to compete with each other, with customers buying family security services or residential fire insurance separately. This study proposes that they should work together to create mutually beneficial tactics in view of cooperative-competition theory and the real-life situations based on risk and financial management.
As the improvement of financial performance in the security service industries depend on the normal expansion of business, the security industry should promote decision analysis applications to deal with the uncertainty of consumer demand going forward. It is foreseeable that the competition will intensify with competitors from other sectors. Kim et al. (2013) found that academic literature has emphasized cooperation between channel members because of the interdependence between them; in reality, retailers may accept competition as just another part of doing business with suppliers. This paper viewed the family insurance services as retailers and residential fire insurance as suppliers on product integration, not only to enhance the effect of consumer’s risk aversion, but also intensifies the competitiveness of the industry. This paper aims to examine the decision analysis pertaining to the integration of the residential security products and fire insurance policies to meet the demand from risk-averse consumers. The purpose is to calculate and validate the appropriate discount of insurance premiums that insurance companies provide to security companies for the combined product offerings. This is followed with the estimates of the market share for the residential security industry with the enhanced benefits to risk-averse consumers by combining risk control into the fully integrated services.
This paper specifically explains that the discount rate ( ) is not the capital cost rate or return on investment defined under general actual business activities, but the modified discount rate considering the perpetual operating value, which will be adjusted vary with operating costs of the relevant industries. The increase of the written premiums for the residential fire insurance can also be estimated and compared with the different decision analysis results between the Taiwan and London security markets from the viewpoint of sustainable value. The London market is compared with the Taiwan market because that Taiwan’s security market is developing rapidly and the experience of London security market in the UK is quite mature and worth learning. Therefore, the author uses the model mentioned before, alongside practical data, to illustrate this comparison, mainly to indicate the feasibility of this model and to verify its consistency and the differences of the results in a multinational market.
(P20)
This paper derives findings with POLYMATH calculations. To estimate the number of customers required for the economic scale ( ) and the appropriate premium discount ( ) involved the sustainable value. And as the risk control measures of the private security company, Eq. (2) estimates the market share of the family security service. The result suggests about 1.36 times (=0.2449/0.18). may be set into Eq. (1), so this paper can estimate the amount of written premium on residential fire insurance going forward. In view of the product-service system as a new model, the integration of goods and services provides customers with added sustainable value and lays the cornerstone of competitive strategy (See Phumbua and Tjahjono 2012). This paper presents the five main different results between Taiwan and London with ten different parameters of the family security market to identify the optimal number of family security companies offering integrated service. The number for London is about 1.349. The improvement of the risk aversion effect will boost the market share of the family security services offering integrated products from 18% to 24.49% in London, and from 3% to 11.62% in Taiwan.
- And to clearly state the mechanisms and specific policies through which own findings can serve as reference for decision making, not only simply stating this.
Response:
(P20)
In response to consumers' needs of integrated hedging, it is clarified in this study that the feasibility of specific cooperation policies between the insurance and the security industries can be verified through mathematical model establishments and reasonable data analyses. By importing actual data, estimated results are generated, which can be used as a reference for decision-making in business marketing to achieve sustainable value operation.
- Especially, in the conclusions section, instead of restating what the authors did in their paper, which is already mentioned by that section too many times, the authors should start a debate on the results (a section which is practically missing) and the subsections of academic and managerial implications, as well as research contributions should be significantly enhanced.
Response:
In the conclusion section, the author removes the repetitive content in the paper and discusses its results instead, in response to the review suggestions, as the following:
(P21)
- Based on practical experiences in insurance and security services, the author establishes a mathematical model to obtain a reasonable discount result, which can be used as a reference when two parties cooperate and negotiate with each other.
- This article is about the application of sustainable value in the service industry. The consumer-integration services provided via a cooperation between insurance and security service companies will meet consumers' more comprehensive risk and financial planning and will enhance the insurance and security service companies’ business market share. This will be verifiable in practice.
- The integration mechanism of the insurance and the security industries can be applied to online alliance marketing services, allowing consumers to shop online through the Internet of Things. In this way, sustainable industrial operation could increase discounts on the items sold online to reciprocate consumers.

Reviewer 2 Report
Dear Autors,
I am sending comments on the article in the review.
Best regards,

Author Response
Thank you for your opinions (reviewer 2), the author responded as follows:
- Research method.
The authors built a mathematical model based on 10 parameters related to the London insurance market, and the analysis covers as many as 36 variables that are presented on 3 pages of the article. The number of variables presented in the article, accepted for analysis within the model, is too long, and as a result, the analysis is difficult to read. In this article, consider limiting the number of analyzed variables.
Response:
(P6)
Because the handling of risks involves the business revenue and expenditure results of two different industries, there are related factors involved in financial analyses with a single tool and with an integrated tool. In order to perform a complete comparison and discussion, there are more parameters to consider in this study than other general research articles. For the parameter estimations and calculations based on the industries’ meeting consumers’ varied aspects of needs for hedging, the author re-organized the parameters into seven categories under the same group function to facilitate the review of this paper. In addition, a sensitivity analysis is conducted after the model construction to reduce the natural bias caused by importing too many parameters into the model.
(P6) Written premium related-
(P7) Market share of family security services related-
Private security companies related finance-
(P8) Finance of integrated services related-
(P9) Premium discount related-
Loss payouts of the security companies related-
One-time investment by the private security companies-
- Discussion.
The authors inform that the analysis concerns the markets of London and Taiwan. Unfortunately, the article does not justify why the two markets were selected for comparison, to what extent they are similar and how different. Taiwan is a country with a population of approximately 24 million, covering an area of 36,193 km2, and London is a city of approximately 8.9 million inhabitants and an area of 1,572 km2. The markets are likely to be different, so how do the authors make the comparisons? This requires an explanation and justification.
Response:
(P2)
The London market is compared with the Taiwan market because that Taiwan’s security market is developing rapidly and the experience of London security market in the UK is quite mature and worth learning. Therefore, the author uses the model mentioned before, alongside practical data, to illustrate this comparison, mainly to indicate the feasibility of this model and to verify its consistency and the differences of the results in a multinational market. Another reason is that the London market is the most representative one in the world in terms of its scale or legal system. Its market information is relatively complete. Thus, it is worthy to use the London market as a contrast group for the model’s development.
The authors provide information about the UK insurance market, but there is little information from an article about the Taiwan insurance market, other than that "Taipei government's requirement to install firealarms in 2017 ”. Does this requirement apply to all cities in Taiwan? The London and Taiwanese markets are culturally different, they probably operate on the basis of different legal regulations, are characterized by different degrees of maturity, so what assumptions do the authors base about creating the model? It also requires an explanation and justification.
Response:
(P5)
In the research article, the author specifically mentioned that the Taipei City Government’s requirement of installing fire extinguishers in 2017 proves that they pay great attention to fire accidents. Whereas, due to urban-rural differences, other cities do not have this requirement. The author imports relevant data of the London and the Taiwan market to the basic model. In theory, the data of the London market is mainly used as a contrast group. While in practice, the feasibility of the model can also be verified in different markets and scales.
- References.
The bibliography should be organized. Please check the surname and first name of the author of the publications listed. The authors of the article confuse this and have a problem with the alphabetical ordering of items (for example: Brigitte, Hoogendoorn, Peter, van der, Zwan and Roy, Thurik. 2019 should be listed under the letter H. (Hoogendoorn Brigitte, ....)
Response:
This article will be reconfirmed and adjusted to a consistent reference format in accordance with the reviewer’s suggestions and the journal’s relevant format regulations.
(P23-24)
Hoogendoorn, Brigitte, Zwan, van der Peter and Thurik, Roy. 2019. Sustainable Entrepreneurship: The Role of Perceived Barriers and Risk. Journal of Business Ethics 157:1133–54.DOI 10.1007/s10551-017-3646-8.
Huber, Peter and Nowotny, Klaus. 2020. Risk aversion and the willingness to migrate in 30 transition countries. Journal of Population Economics 33:1463–98.
https://doi.org/10.1007/s00148-020-00777-3.
Kim, Stephen, Kim, Namwoon, Pae, Jae H. and Yip, Leslie. 2013. Cooperate
and compete: Coopetition strategy in retailer-supplier relationships. Journal of
Business & Industrial Marketing 28(4):263-75.
Rahman, Aida Lope Abdul Alifah, Islam, Shareeful, Kalloniatis, Christos and
Gritzalis, Stefanos. 2017. A Risk Management Approach for a Sustainable Cloud
Migration. Journal of Risk and Financial Management. 10, 20-38.
doi:10.3390/jrfm10040020
Štrukelj, Tjaša, Nikoli´c, Jelena, Zlatanovi´c, Dejana and Zabukovšek, Sternad Simona. 2020. A Strategic Model for Sustainable Business Policy Development. Sustainability 12(2): 526-53. https://doi.org/10.3390/su12020526